# Caddisflies (Trichoptera) of Protected Calcareous Fen Habitats: Assemblages, Environmental Drivers, Indicator Species, and Conservation Issues

**DOI:** 10.3390/insects14110850

**Published:** 2023-10-31

**Authors:** Edyta Buczyńska, Adam Tarkowski, Piotr Sugier, Wojciech Płaska, Andrzej Zawal, Anna Janicka, Paweł Buczyński

**Affiliations:** 1Department of Zoology and Animal Ecology, University of Life Sciences in Lublin, Akademicka 13, 20-033 Lublin, Poland; edyta.buczynska@up.lublin.pl; 2The University Centre for Environmental Studies and Sustainable Development, University of Warsaw, Żwirki i Wigury 93, 02-089 Warsaw, Poland; adam.tarkowski@uw.edu.pl; 3Department of Botany, Mycology and Ecology, Institute of Biological Sciences, Maria Curie-Sklodowska University, Akademicka 19, 20-033 Lublin, Poland; piotr.sugier@mail.umcs.pl; 4Department of Hydrobiology and Protection of Ecosystems, University of Life Sciences, Dobrzańskiego 37, 20-262 Lublin, Poland; wojciech.plaska@up.lublin.pl; 5Centre of Molecular Biology and Biotechnology, Institute of Marine and Environmental Sciences, University of Szczecin, Wąska 13, 71-415 Szczecin, Poland; andrzej.zawal@usz.edu.pl; 6Department of Zoology and Nature Protection, Maria Curie-Skłodowska University, Akademicka 19, 20-033 Lublin, Poland; anna.janicka14@gmail.com

**Keywords:** caddisflies, wetlands, astatic waters, drying out, bioindicators, area protection, conservation

## Abstract

**Simple Summary:**

Calcareous fens, a unique and endangered type of peatland, harbour unique groups of little-studied aquatic insects. Caddisflies (Trichoptera), an amphibiotic insect group, have proven to be useful (at different levels of organisation) in describing various aspects of fens and their water bodies (pools and ditches). We focused on the evaluation of indicator species and the most important patterns and drivers of species distribution. A comprehensive approach including species, assemblages, functional groups, and ecological indices was recommended. In general, our findings provide a solid basis for analysing the potential of caddisflies in such habitats and present some useful tips for conservation practices and the management of these vulnerable ecosystems.

**Abstract:**

The caddisflies (Trichoptera) of calcareous fen habitats, in contrast to those of other peatland types, have been poorly researched. We thus conducted a two-year study in south-eastern Poland encompassing four types of such habitats—drained and undrained fens and water bodies (pools and ditches) located within the fens—in order to define trichopteran reference assemblages (PCoA), indicator species (IndVal analysis), and the drivers (both natural and those associated with landscape management, including area protection) responsible for caddisfly species distribution (CCA). The most important environmental driver was habitat persistence. Distance-based RDA analysis revealed a distinct pattern in the distribution of species with or without diapause along the persistence gradient. Environmental drivers associated with plants were also crucial for both fens and water bodies. The key factor influencing the caddisfly assemblages of pools and ditches was the use and management of the surrounding land, whereas in the fens, it was the level of area protection. Physical and chemical water parameters had no statistically significant impact on the assemblages. Some factors can be modified by humans (e.g., water level regulation, vegetation, and landscape management) to maintain healthy ecosystems for aquatic insects.

## 1. Introduction

Peatlands, encompassing peat bogs and fens, are amongst the most endangered hydrological ecosystems in the world [1]. In Europe, acidic peat bogs dominated by *Sphagnum* are well-defined and fairly uniform, with distinctive and well-known flora and fauna. However, fens have been classified in widely differing ways; their constant and best-researched botanical and hydromorphological components do not allow a full, ideal definition of these habitats. It can generally be assumed that brown mosses, sedges, grasses, and trees (*Betula*, *Salix*, and *Alnus*) are typical of fens, in contrast to bog peatlands, which are dominated by *Sphagnum* mosses and ericaceous shrubs [2]. The key features of fens are the pH and nutrient gradients, which give rise to a habitat continuum ranging from low-nutrient acidic environments to nutrient-rich alkaline environments. In addition, in some fens, there are habitats that are always wet and habitats dry out periodically, whereas others may or may not contain anthropogenic or animal-constructed water bodies. Among the nutrient-rich alkaline environments, the calcareous fens at one end of the pH/calcium gradient are a highly specific habitat; in Poland, they are unique because of their origin (surface karst processes), the chemical parameters of the water, and the extensive fen sedge beds (*Cladium mariscus* (L.) Pohl) that they support.

Fens are generally rich in species of high natural value and have a narrow ecological amplitude [2,3,4]. In Poland, 92.4% of all peat bogs are fens [5], so for this reason alone, it might seem that they should have already been well-explored. While the vegetation forming the basis of their typology has been adequately identified, little is known about the many groups of animals inhabiting them, especially invertebrates. In Poland, the best-known insects in sedge-moss alkaline fens are Odonata, Lepidoptera, and Coleoptera; by contrast, very little is known about other orders [6]. We know even less about the relationships between particular species and natural and anthropogenic habitat factors, especially on larger spatial scales [7,8]. In the case of peatlands, there are papers on aquatic invertebrates (whole orders or certain groups), but solid data on the relationships between caddisflies and environmental factors are practically non-existent. Key factors for peat pools in *Sphagnum* peat bogs were found to be dissolved oxygen, *Sphagnum* cover, and pool perimeter [9], but corresponding data for fens are not available. This also holds for Trichoptera, an order of amphibiotic insects which, as recognised indicators of the state of rivers or sources (e.g., [10,11]), could also serve as a means of assessing the natural value (reference states) of these sensitive habitats and detecting the changes that they are subject to. Caddisflies (Trichoptera) are a species-rich order with wide-ranging habitat preferences [12,13]. They are intimately associated with aquatic and terrestrial vegetation [9,14] and could make a valuable contribution to the already rich store of botanical knowledge of fen habitats. Trichopterans appear to be a promising group for studying fen sites containing a range of water bodies of varying vegetation, size, and permanence [7]. This is crucial because not all aquatic insects can tolerate the periodic drying out of their habitats, and some groups occur exclusively in deeper fen water bodies or, conversely, in sites devoid of these. Caddisflies thus inhabit the entire spectrum of fen habitats, which translates into a more accurate picture of the trichopteran fauna and its species richness, including the presence of potentially endangered taxa [2,6]. 

In Europe, many fens have been drained and converted into agricultural land, and the estimated losses of this type of habitat in several countries can reach 98% [1,15]. Fen remnants, constantly subjected to human pressure and further exposed to adverse natural factors, are rapidly deteriorating. The chief factors responsible for the degradation of fen habitats are drainage, changes in regional hydrology, eutrophication, abandonment of traditional land use, and water extraction [3]. In Poland, even though fens make up a huge percentage of peat bogs, we may lose them before we are able to understand them properly, especially regarding their invertebrate fauna. Although many of the most valuable fen habitats, including calcareous fens, are protected, we cannot be certain that this will be sufficient and to what extent this protection actually extends to aquatic invertebrates. Crucially, in the case of fens, which are important Natura 2000 habitat types (Nos. 7150 and 7210—priority habitats—and No. 7230), no species or reference groups of Trichoptera have ever been designated that could form the basis of monitoring programmes for these declining and endangered habitats. The same applies to indicator or umbrella species. So far, only *Hagenella clathrata* (Kolenati, 1848) has been identified as a typical umbrella species for peat bog waters and wetlands in Poland [16], but this is a species associated with raised and transitional (*Sphagnum*) bogs. 

In view of the above, our research addressed the following questions. (i) How do the trichopterans of drained and undrained fens and of the nearby artificial watercourses and water bodies function at different levels of organisation? (ii) Which factors relating to fens and their utilisation and protection are important for their trichopteran fauna, and how can we manage and protect them? (iii) Do the forms of conservation affect the trichopteran fauna, and if so, how? (iv) Which species can be regarded as references and indicators for calcareous fens (including Natura 2000 habitat code—7210).

## 2. Materials and Methods

### 2.1. Study Area and Research Sites

The research was carried out in western Polesie, a vast (c. 130,000 km^2^) plain stretching from Poland across Belarus and Ukraine into western Russia. Although it is mostly a farming landscape, there are large expanses of forest and semi-natural ecosystems. Its subterranean waters lie at shallow depths and have given rise to numerous water bodies and wetlands, which make up more than half its overall area [17,18].

We studied two macroregions: western Polesie (sites FP1-FP3) and Wołyń Polesie (the other sites) [19]. The karst and thermokarst processes that took place in the chalky substrate led to the formation of drainless hollows, in which peat [17] and, subsequently, distinctive calcareous fens formed. Rich in substrate-derived calcium, their only source of moisture is rainwater (the chalky substrate is impermeable); moreover, they are mesotrophic, slightly alkaline, and always moist, although the water levels fluctuate annually. The dominant vegetation consists of calciphilous emergent plants, principally *Cladium mariscus* (L.) but also *Carex buxbaumii* Wahlenb. and *Schoenus nigricans* L. This is a priority habitat in the Natura 2000 programme (“7210 Calcareous fens”), which, in Poland, is strongly regressing and threatened because of, among other factors, its very small overall surface area and the tiny area of some sites, many of which are widely scattered [20,21]. In other countries in the EU, such habitats are also widespread, and their conservation status is inadequate or bad, except in the northern part of the Baltic Sea basin, in Hungary, the Czech Republic, and locally in Austria [22].

A total of 21 sites were explored. They reflected the diversity of aquatic and wetland habitats among the calcareous fens due in large measure to the human influence on these areas (Figure 1). 

Undrained fens: UF-1 (51.181665 N 23.524231 E), UF-2 (51.167406 N 23.605197 E), UF-3 (51.156056 N 23.591528 E), UF-4 (51.138361 N 23.685722 E), and UF-5 (51.162167 N 23.658028 E). These patches of fens never drained, without drainage ditches or channels. Water levels varied from falling but with water always covering the whole surface of a site to drying out, though never completely. At site UF-5, the water level never changed because its drainage was blocked by a railway embankment. Dense emergent vegetation was dominant, mostly *Cladium mariscus* and/or *Phragmites australis* (Cav.) Trin ex Steud., often with substantial admixtures of *Carex* spp. and, locally, *Molinia caerulea* (L.) Moench. Only at site UF-4 did the dominant emergent vegetation consist of clumps of *Carex elata*. Permanently waterlogged substrate supported large patches of *Chara intermedia* A.Braun. Site EF-4 was fish-free, but the following species were recorded at the others: *Carassius gibelio* (Bloch, 1782) (UF-1, UF-2, and UF-5), *Perccottus glenii* Dybowski, 1877 (UF-3 and UF-5), *Perca fluviatilis* L. (UF2), and *Rhynchocypris percnurus* (Pallas, 1814) (UF-1).

Drained fens: DF-1 (51.163945 N 23.501003 E), DF-2 (51.173695 N 23.534203 E), DF-3 (51.180477 N 23.576393 E), DF-4 (51.146000 N 23.639333 E), DF-5 (51.137322 N 23.682622 E), and DF-6 (51.139222 N 23.702889 E). These were similar to those in the undrained group but with drainage ditches. Water persisted on the fen surface until May (DF-6), June (DF-1, DF-2, and DF3), or mid-July (DF-5); in one case, it reappeared in the autumn (DF-2). Only at DF-4 were there small patches of fen that did not dry out entirely. The emergent vegetation was dominated by *Carex* spp. associations with admixtures of *Phragmites australis*, sometimes with clumps of *Cladium mariscus*, which were still numerous at DF-2 and DF-5 or strongly regressing at DF-4. Only at DF-6, utilised as a mown meadow, was *Molinia caerulea* the predominant species, which together with *Phalaris arundinacea* L. made up a signific admixture at site DF-3. Fish (*Carassius gibelio*) were present only at DF-1.

Fen ditches: FD-1 (51.163879 N 23.501536 E), FD-2 (51.180269 N 23.576946 E), FD-3 (51.181098 N 23.576292 E), and FD-5 (51.139441 N 23.703683 E). (1) Large, first-order ditches with steep banks, partially concreted over, c. 4 m wide and c. 1 m deep, with a visible flow of water, and periodically dredged (FD-2, FD4); (2) second-order ditches, similar to the first-order ones, except that the water was stagnant for much of the year (FD-3); (3) a small (width up to 1 m), overgrowing, and disappearing ditch on the fen (FD-1). These ditches always contained water, their banks supported *Phragmites australis*, which was replaced by *Oenanthe aquatica* (L.) Poir. in the middle of the ditch at FD-2; at FD-3, the vegetation consisted of *Sparganium erectum* L. em. Rchb., *Molinia caerulea*, and *Typha latifolia* L. There were no floating plants, and submerged ones were present only at sites FD-2 (*Elodea canadensis* Michx.) and FD03 (*Hottonia palustris* (L.) Moench, *Myriophyllum verticillatum* L.). Although there were very large fluctuations in the water level in FD-1, it never dried out completely. The emergent vegetation there consisted of *Carex elata* with clumps of *C. acutiformis*, and small patches of *Lemna minor* L. were growing on what little open water there was. *Carassius gibelio* was present at all these sites, and *C. carassius* was also present at FD-4.

Fen pools: FP-1 (51.365993 N 23.107001 E), FP-2 (51.365598 N 23.106813 E), FP-3 (51.365642 N 23.105059 E), FP-4 (51.173766 N 23.534562 E), FP-5 (51.157083 N 23.591556 E), and FP-6 (51.144210 N 23.632724 E). Dug manually, these pools were old, small (0.001–0.27 ha, av. 0.08 ha), shallow (depth 1–1.5 m), with vertical banks, permanent, with brown though transparent water, and dy-type sediments on the bottom. They lay in open terrain; only the pools at Garbatówka (FP-1, FP-2, FP-3) partially bordered alder swamps. The emergent vegetation was poorly developed at FP-1 and FP-2, moderately so at FP-3, and strongly at FP-4, FP-5, and FP-6. This consisted of a mixture of sedge fields and components of tall emergent vegetation (*Phragmites australis*, *Typha* spp.), locally with admixtures of *Equisetum fluviatile* L. (FP-1) or *Alisma plantago-aquatica* L. (FP-6). Only at FP-5 was *Cladium mariscus* dominant. The submerged vegetation consisted of well-developed *Charetea* meadows (FP-1, FP-2, FP-3, and FP-4). At these same four sites, there was floating vegetation, *Stratiotes aloides* L. being abundant at the first three; apart from this, there were *Potamogeton natans* L. (FP-1, FP-2, FP-3, and FP-4), *Nuphar lutea* (L.) Sibth. & Sm. (FP-1 and FP-3), and *Nymphaea alba* L. (FP-4). The fen pools were the richest in fish of all the habitats explored, with *Carassius gibelio* (FP-2, FP3, FP4, FP-5, and FP-6), *Rhynchocypris percnurus* (FP-1, FP-2, FP-3, FP-4), *Carassius carassius* (FP-1, FP2, FP4), *Perccottus glenii* (FP-4, FP-5, FP-6), *Perca fluviatilis* (FP-2), and *Rhodeus sericeus* (Pallas, 1776) (FP-1).

### 2.2. Field and Laboratory Procedures

The material was collected once a month from April to October (except August) in 2016 and 2017. Aquatic stages of caddisflies were caught with a 1 mm mesh square-framed pond net from an area of c. 2.5 m^2^ in the first year of fieldwork and 5 m^2^ in the second. All the microhabitats in a particular water body or on a patch of fen were sampled. The material was packed into plastic bags and transported to the laboratory, where it was sorted in a sampling tray. All the caddisflies found were conserved in 70% ethanol. Next, they were identified at the species level according to the taxonomic keys of Edington and Hildrew [23], Wallace et al. [24], and Rinne and Wiberg-Larsen [25].

During each sampling bout, the properties of the water were measured in situ with a Hanna Instruments HI 9828 device, including temperature (TEMP) (°C), pH, oxidation reduction potential (ORP) (mV), dissolved oxygen concentration (DO) (mg·dm^−3^), electrolytic conductivity (EC) (μS·cm^−1^), and total dissolved solids (TDS) (ppm). The differences in all these parameters except pH between the various habitats were highly statistically significant (Table 1). In addition, the following characteristics of each site were assessed: permanence/astatism (expressed as the number of months during which water was present at a site) and the overall % coverage of a site by vegetation. The coverage of the following was also assessed: emergent vegetation, floating plants (fen pools and ditches), underwater meadows, and the coverage of a site by trees and shrubs. These data were scored from 0 (absence/undeveloped) to 4 (complete cover/strongly developed) for the statistical analyses. In addition, the conservation status was scored as follows: nature reserve—5, landscape park—4, protected landscape area—3, Nature 2000 area—2, and ecological ground—1; the overall result for a particular site was the sum of these scores. Land use was scored by counting the presence of roads, water, fens, meadows, fields, and forests (maximum score = 6). The mowing of each site was also taken into account, and once a year in the early summer, its coverage by plant associations was assessed using the Braun-Blanquet method [26].

The occurrence of fish at each site was analysed. This fieldwork was carried out using an impulse current fishing module (IUP-12; 220–250 V, 7A). A single fishing electrode was used while wading through the water, as the littoral zones of these water bodies were no deeper than 1 m. Characteristic habitats were earmarked for this fieldwork, the fishing zone was usually 100 m long, and fish were caught within a radius of 2 m from the shoreline. All the fish caught (except for *Perccottus glenii*, which, according to the Regulation of the Minister of the Environment [27], was killed with Morenicol Lernex Sedation Fluid and then disposed of) were later returned unharmed to the habitat from which they were taken. The number of species caught at a particular site was used in the statistical analysis.

### 2.3. Data Analyses

The following ecological indices were used to analyse the caddisfly fauna: species richness (S), abundance (N), the Shannon–Wiener diversity index (H), the Buzas–Gibson evenness (E), the Simpson dominance index (D), and the estimate of total species richness (Chao1). The Jaccard and Bray–Curtis formulas were used to calculate faunistic similarities between the four types of habitats.

Species representing two ecological groups were analysed in the context of the habitat specificity associated with the site persistence and their sensitivity to climate change: (1) taxa with varied drought-resistance strategies [28]—species with adult diapause and species without diapause—and (2) climate change vulnerability scores (CCVS) [29]. The first functional group was also involved in the distance-based RDA (db-RDA) method based on Bray–Curtis dissimilarities showing the effect of the “persistence” factor on species representing the particular categories. 

Principal coordinates analysis (PCoA) with square-root transformation and the Bray–Curtis index were used to characterise the faunistic similarities between sites representing a particular fen type and the distribution of caddisfly species. Data on climate change vulnerability scores (CCVSs) for the species collected were also incorporated into this method.

Non-metric multidimensional scaling (NMDS) using the Bray–Curtis formula was used to highlight faunistic patterns between sites subject to different forms of conservation (nature reserves and Nature 2000 areas).

To identify the most important environmental drivers for trichopteran species, canonical correspondence analyses (CCAs) with the forward selection (FS) procedure (999 test permutations) were used. Separate analyses were run for the two main habitats: fens (drained and undrained) and water bodies (pools and ditches), including three sets of variables for each habitat: (1) physical and chemical water properties, (2) parameters associated with aquatic and terrestrial plants, and (3) factors associated with fish, protection, land use and site persistence. CCA biplots indicated the variables primarily responsible for species variation.

Using the similarity percentage method (SIMPER) [30], we identified the species responsible for the faunistic dissimilarity (Bray–Curtis matrix) between the two main habitats: fens and water bodies located within the fens. We also performed indicator species analysis IndVal [31], which yielded the indicator species for both types of sites. In this analysis, statistical significance (*p* < 0.05) was estimated by 9999 random permutations of sites across groups.

PCoA, CCAs, and distance-based RDA were performed in Canoco 5.0 [32], and NMDS, Simper, and IndVal were performed in the PAST 4.13 program [33].

The Mann–Whitney U test was used to compare two independent samples, and the Kruskal–Wallis was used to compare a larger number of samples. The statistical significance threshold was *p* = 0.05. The computations were performed in Statistica 13.0. The box plots were also generated in this program. 

## 3. Results

### 3.1. General Characteristics of the Trichopteran Assemblages and Ecological Groups

A total of 1065 specimens of the aquatic developmental stages of Trichoptera were obtained. Fifty per cent of them were caught in the fen ditches and twenty-seven per cent were caught in the fen pools, whereas the numbers caught in the two types of fens were very similar. The largest numbers of species were recorded in the fen pools and drained fens—10 each—but only about half as many were recorded in the ditches and undrained fens—6 and 5, respectively (Table 2). The differences between the two indices calculated per sample were considerable but not statistically significant (Figure 2). The large number of specimens obtained did not translate into a large number of species. This was best exemplified in the ditches, in which the dominant species was *Limnephilus flavicornis* (F.) (98% of all the specimens), and also in the drained fens, where the abundance was low but the species richness was high. The only species occurring in all four habitat types, which was also the most numerous one, was *L. flavicornis*. The highest numbers of exclusive species were obtained in the fen pools and drained fens, with four each, and one each was found in the ditches and undrained fens. Comparing fens with water bodies, we found 12 and 13 species, including 5 that were exclusive to each. By contrast, the abundance was over three times higher in water bodies than in fens.

Among the alpha diversity indices of all the habitats, dominance (D) was the lowest in the drained fens (D = 0.31) but was relatively high in the other habitats (fen pools—D = 0.82, undrained fens—D = 0.88), reaching a maximum in the fen ditches (D = 0.98). The Shannon–Wiener diversity index (H) was the highest in the drained fens (H = 1.5) and the lowest in the fen ditches (H = 0.1), while the values for the fen pools and undrained fens were H = 0.5 and H = 0.3, respectively. The Buzas–Gibson evenness (E) was the highest in the drained fens (E = 0.44), followed by the undrained fens (E = 0.27); the values in the fen ditches and pools were similar (E = 0.16 and E = 0.18). The Chao1 estimate of total species richness was the highest for the water bodies: fen ditches—Chao1 = 12 and fen pools—Chao1 = 11.5; drained fens had a value of Chao1 = 10, and the undrained fens had a value of Chao1 = 6. Bray–Curtis similarities (S) for the four habitat types were the highest between the pools and ditches—S = 64%—and between the pools and undrained fens—S = 62%; drained fens were the least similar to the ditches—S = 15%. 

Jaccard’s faunistic similarities showed that there were two groups of most similar habitats: fen pools and undrained fens (36%) and fen ditches and drained fens (33%). Particular sites representing specific habitats revealed the following pattern: fen pools and ditches occupied the left side of the PCoA diagram (except for FP_5) but on the opposite side of axis 1 (Figure 3). Species more sensitive to climate change were also dominant in this part of the diagram. In turn, fens (both types, but not UF_1), together with species not sensitive to climate change, were placed on the opposite side, signifying that axis 1 is a habitat–climate gradient. Axis 2, in turn, is largely a site persistence gradient; except for *Athripsodes aterrimus* (Stephens, 1836), typical for the ditches, all the species in the lower half of the diagram enter diapause. The upper left part of the diagram represents fen pools with permanent water species, while on the right, there is a mixed group, but it is wholly insensitive to climate change. PCoA also enables reference species to be selected for the various types of habitat: fen pools—*Agrypnia varia* (F.), *Triaenodes bicolor* (Curtis, 1834), *Erotesis baltica* McLachlan, 1877, *Holocentropus picicornis* (Stephens, 1836), *Oecetis furva* (Rambur, 1842), and *Limnephilus flavicornis*; ditches—*Athripsodes aterrimus*, *Grammotaulius nigropunctatus* (Retzius, 1783), *Limnephilus flavicornis*, and *L. subcentralis* Brauer, 1857; and fens—*L. griseus* (L.), *L. auricula* Curtis, 1834, *L. binotatus* Curtis, 1834, *L. fuscicornis* (Rambur, 1842), and *Colpotaulius incisus* (Curtis, 1834).

The drained fens were the only habitat occupied by species with adult diapause (Figure 4A). More than 80% of the species caught in the ditches also belonged to this category. In the other habitat types, species with and without diapause were distributed almost evenly, the latter mostly being found in the fen pools. In the climate change vulnerability category, species insensitive to climate change were dominant in the drained fens, whereas more sensitive species were dominant in the fen pools (Figure 4B). In the other two habitat types, the proportion of sensitive and insensitive species was much the same; in the ditches, it was 50:50, but sensitive species prevailed in the undrained fens.

### 3.2. Environmental Factors Influencing the Trichopteran Assemblages

The measured physicochemical parameters did not influence the occurrence of caddisflies in any of the studied habitats. These parameters explained 14% of the overall variation in Trichoptera in the fens but a mere 9.2% in the water bodies. 

However, the analysis of the vegetation structure yielded different results; for the fen caddisflies (Figure 5A), the most important factor was the density of the emergent vegetation (responsible for 35% of the total variation, *p* = 0.016). Species like *Limnephilus fuscinervis* (Zetterstedt, 1840) were most closely associated with dense patches of helophytes, whereas *L. flavicornis* and *Oecetis furva* occurred at sites with a much looser emergent plant structure. The overall % vegetation coverage of a site also contributed substantially to explaining the total variation in caddisfly occurrence in the fens (21%), albeit at *p* = 0.066. Again, among the five vegetation parameters defined for the fen water bodies, tree coverage of the site was responsible for 32.5% of the total variation (*p* = 0.021). Positively correlated with this parameter was a large group of species, i.e., *Limnephilus subcentralis*, *Erotesis baltica*, *Agrypnia varia*, *Oecetis furva*, and *Holocentropus picicornis*, which preferred sites with plenty of trees. *Limnephilus flavicornis* was associated with intermediate values of this factor, but *L. griseus* displayed a strong affinity for open areas. The variables associated with aquatic and terrestrial vegetation explained 13.2% of the total variation of caddisflies in this type of habitat.

The variables associated with fish, protection, land use, and site persistence turned out to be the most significant for the caddisflies of the fens, as they explained 32% of the total variation (Figure 5C); for the fen pools and ditches, however, the same variables explained 12.2%. The key factors of the fens were site persistence (responsible for 42% of the total variation, *p* = 0.001) and the degree of area protection in a particular site (correspondingly 23% and *p* = 0.05). The values of the two other parameters lay slightly above the significance threshold, with *p* = 0.069 (land use score) and *p* = 0.07 (pressure on the part of fish). The only key factor for the caddisflies of artificial water bodies was the extent to which the surrounding land was utilised/managed (*p* = 0.004), which made a 41.5% contribution to the total variation (Figure 5D). 

In the case of the fens, the positive correlations regarding the two most important variables applied to only a few species, and their magnitudes were average for *L. flavicornis* and *Trichostegia minor* (Curtis, 1834) and slightly higher for *Limnephilus binotatus* (this species can be treated as being associated with the most carefully protected sites). *L. auricula*, *L. griseus*, and *Grammotaulius nigropunctatus* lay on the opposite side of the site persistence vector, which testifies to their association with astatic water bodies. In the fen ditches and pools, five species (*Agrypnia varia*, *Oecetis furva*, *Holocentropus picicornis*, *Triaenodes bicolor*, and *Erotesis baltica*) were strongly associated with sites with a higher surrounding land use/management index, while *Limnephilus flavicornis* had intermediate values of this variable.

Overall, the CCA results for the fens alone showed that 14 of the parameters explained 67% of the variation in caddisflies (three key parameters, with another three in the higher significance interval) but slightly over half that value was observed for ditches—34.6%—and only two parameters were key ones.

Since the most important parameter in all the above analyses was site persistence, db-RDA was carried out for the entire assemblage of fen trichopterans, taking into account this single variable and one functional group criterion, i.e., the reaction of caddisflies to drought, expressed as the occurrence or non-occurrence of diapause. The results (Figure 6) showed that this parameter is key to the distribution of caddisflies from a specific category in the form of a strong positive correlation with its vector, indicating non-diapausing species, i.e., *Agrypnia varia*, *Oecetis furva*, *Holocentropus picicornis*, *Triaenodes bicolor*, and *Erotesis baltica*. For their development, they require the uninterrupted presence of water all year. *Trichostegia minor*, *Colpotaulius incisus*, *Limnephilus stigma* Curtis, 1834, and *Limnephilus fuscicornis* were placed in the centre of the plot; in the calcareous fens, these species could tolerate brief periods of drought. *Limnephilus auricula*, *L. griseus*, and *L. binotatus* were the most drought-resistant species; on the plot, they occupied the opposite position relative to the vector of the variable. All the species not diapausing in summer were on the left side of the plot and were thus positively correlated with the vector. *Limnephilus subcentralis* lay close to the large group of non-diapausing species, like *Limnephilus flavicornis* and *Grammotaulius nigropunctatus*, situated slightly farther away; these species, despite diapausing, have a broader range of tolerance and can colonise permanent waters equally well.

Mowing significantly influenced the abundance of *Grammotaulius nigropunctatus* only (*p* = 0.025). This species evidently preferred unmown sites. For all the other species and all the ecological indicators, this factor was not significant.

### 3.3. Protected Fens, Reference and Indicator Species

The layout of the study sites by protection status (Figure 7) revealed certain regularities. All Natura 2000 sites (dots) lay along or on axis 1 (except FP-4 and DF-2) in the lower half of the plot. The sites outside Natura 2000 areas (squares) formed two blocks. The first, representing ditches, lay at the very end of axis 1, whereas the second, covering the drained fens, contained the more distant sites (with a lesser faunistic similarity) and was placed in the right upper quarter. Nature reserves (shown in red) were also distributed in pairs along the first axis, whereby caddisfly assemblages in FP-4 and UF-1 were the most similar and DF-2 and UF-5 lay farther away, though closer to the sites corresponding to their types. This arrangement clearly indicates that coordinate axis 1 is evidently a conservation gradient associated with its classification as a Natura 2000 area, while coordinate axis 2 is a habitat-type gradient. With a few exceptions, the left side of the plot represents mainly anthropogenic water bodies (pools and ditches), while the right side represents fens.

Similarity percentage analysis (SIMPER) showed that four caddisfly species were responsible for the differences between the two main habitat types, i.e., fens and water bodies, contributing 90% to the total dissimilarity; these were *Limnephilus flavicornis* (main statistics: average dissimilarity = 47.4, % contribution = 72.5%, mean fens = 4.9, mean water bodies = 15), *L. griseus* (average dissimilarity = 6.4, % contribution = 10%, mean fens = 1.4, mean water bodies = 0.03), *Oecetis furva* (average dissimilarity = 2.4, % contribution = 3.7%, mean fens = 0.08, mean water bodies = 0.2), and *Limnephilus fuscinervis* (average dissimilarity = 2.3, % contribution = 3.5%, mean fens = 0.2, mean water bodies = 0.03). These species represent various ecological elements and have different drought-resistance strategies, but all of them are closely associated with aquatic and emergent vegetation. The results of the IndVal analysis indicated that the best candidates as indicator species for fens were *Limnephilus griseus, Trichostegia minor*, *Limnephilus fuscinervis*, and *Limnephilus stigma*, whereas for water bodies, they were *Limnephilus flavicornis, Triaenodes bicolor, Oecetis furva*, and *Holocentropus picicornis*. *Limnephilus flavicornis* and *Limnephilus griseus* were the most discriminatory pair in the entire fen vs. water body system; this was also confirmed by the statistical significance (*p* < 0.005). 

Among the caddisflies we caught, there were two red-list species, *Erotesis baltica* and *Limnephilus fuscinervis*, representing 11% of all the trichopteran individuals caught. The first one was in the category EX? (probably extinct), and the second one was in the category DD (data deficient) [34]. *Erotesis baltica* was caught in a fen pool, whereas *L. fuscinervis* was present on both the drained and undrained fens and in the fen pools. 

## 4. Discussion

### 4.1. Caddisflies of Calcareous Fen Habitats—Species Richness and Ecological Indices

The aquatic insect fauna of fens is richer than the highly specialised fauna of acidic *Sphagnum* bogs [2]. This also applies to caddisflies, although the overall species richness that we recorded was not especially high (the Chao indices did not indicate a greater expected species richness for particular habitats). For comparison, Kubiak et al. [35] reported 15 autochthonous caddisfly species in the Eppendorfer Moor fen complex in Germany, Hannigan and Kelly-Quinn [36] recorded 20 species in open-water habitats in bogs and fens in Ireland, and Buczyńska and Buczyński [9] found 9 species in peat pools and marginal zones of *Sphagnum* bogs in Poland. However, the results of studies on peatlands covering different habitat types have clearly shown that the invertebrate fauna is richer if the study area includes permanent pools [36]. According to our results, fens and artificial water bodies had a very similar species richness, but the diversity was four times higher in the fens than in the water bodies. Surprisingly, drained fens had twice as many species and an almost fivefold higher biodiversity index than natural fen sites. These were highly distinctive habitats, dominated by species perfectly adapted to desiccation and insensitive to climate changes. In turn, natural undrained fens had the most evenly balanced fauna in terms of functional groups, and their species richness most closely resembled that of rich fen pools. Drainage, though enhancing the faunal diversity, caused its functional structure to become more uniform because only species adapted to extreme conditions were able to survive. Perhaps the lack of competition from species not adapted to desiccation enabled the colonisation of drained fens by the more species that were not sensitive to this factor. Abundance was not a sensitive parameter—it was almost the same on the drained and undrained fens. 

Ecological indices in themselves are insufficient for making a full evaluation of a habitat and applying this knowledge in the planning of restoration activities [37,38]. Here, we need the levels of species and assemblages in the context of species turnover. Were it not for the man-made habitats, five species would be missing from the fauna, and vice versa, restricting the research to fen pools and natural water bodies alone would reduce the general faunistic richness. For species abundance, permanent water bodies were definitely the most important, especially fen pools as refuges. The next level, that of functional groups, also provides valuable information and explanations, which is most clearly evident from the above comparison of the two types of fens. Generally speaking, the faunistic associations among the various habitat types discussed in this paper explicitly demonstrate that habitat heterogeneity, even if resulting from human activities, enhances taxonomic richness. Moreover, when studying the aquatic insects of large fen complexes, especially for inventory purposes or for designating the most valuable sites in an area, it is important that ditches and peat pools are not ignored, even though they are frequently inferior to natural habitats. Besides the now-recognised role of peat pools in peatlands for aquatic invertebrates [8,9,36,37], it is worth recalling the potential of drainage ditches; these may also make a contribution to the landscape-level conservation value [39,40,41,42,43]. The ditches we researched were a refuge for the rheophilic *Athripsodes aterrimus*; this occurred in an assemblage of desiccation-resistant species, which highlights the universal significance of ditches for the trichopterans of both ephemeral and permanent water bodies.

### 4.2. Key Habitat Factors for Trichopterans in the Context of Protection and Conservation

For amphibiotic insects like caddisflies, species preferences involve various spatial scales, from the water itself to the surroundings [10,15,38]. To this, we can add their complex and diverse trophic requirements and the building behaviour of larvae, which is mediated by biotic and abiotic factors. Caddisflies are thus ideal for describing relationships at the water-land levels, in contrast to butterflies and moths, beetles, and snails, which are already well-researched in calcareous fens [6] and usually characterise only the terrestrial aspect of these habitats. The hydromorphological factors and the immediate environment are more important to these insects than the physicochemical parameters of the water. Water was indeed a key factor for caddisflies but in the context of habitat desiccation, which can be evaluated in two ways. On the one hand, this is a negative process, which in the face of an ever-warmer climate [44] may cause calcareous fens to disappear as a consequence of plant succession. The transformation of fens into highly productive, monospecific *Phragmites australis* areas is undesirable [1], and researchers of trichopterans share this opinion. Caddisflies very strongly depend on plants, but different species require diverse plant species; even the structure of the aquatic or riparian vegetation is important [12,14]. Our research demonstrated that even the density of the emergent vegetation at a site is a key factor that must be diversified, i.e., different species require different densities. Importantly, this is a factor that can be actively managed, e.g., by mowing or controlled burning. This is worth doing, as in the case of Natura 2000 habitat No. 7210, the changes in hydrological conditions as well as natural processes like succession and eutrophication are the main threats to calcareous fens [45]. Combatting excessive plant growth can protect the common interests of various groups of animals—not just insects but also birds, like the worldwide endangered aquatic warbler *Acrocephalus paludicola* (Vieillot, 1817), the rarest of migrants in Europe [46]. On the other hand, if only some of the sites in an area dry out periodically, and aquatic insects have permanent refuges in pools and ditches, the species richness increases, and a unique equilibrium is attained, best expressed by the proportion of functional groups in the entire trichopteran fauna. 

Apart from helophytes, the trees and shrubs growing on the banks of anthropogenic waters were also important for caddisflies. This is a significant factor for the feeding preferences of trichopterans, for example. According to Batzer et al. [2], limnephilid caddisfly larvae are important shredders of organic matter (leaves) in peatlands. This group of species, making up as many as 50% of the species we recorded, uses detritus as a food resource, a raw material for building larval cases, and shelter from predators. CCA showed that diverse levels of tree growth suit different caddisfly species. In addition, as in the case of emergent vegetation, this element of the environment can also be appropriately managed in order to maintain particular species of animals and to preserve the greatest possible biodiversity. In general, as Buczyńska and Buczyński [9] also found that in the case of peat pools, structural factors like vegetation, which are easy to control simply by managing the succession, offer interesting prospects for the active conservation of fens and their water bodies. 

Another key factor for caddisflies was the extent to which the surrounding land was utilised and managed. Clearly, it is not only the diversity of study plots on fens but also the extent to which the terrain offers a mosaic of microhabitats fostering aquatic insects that can affect the research results. Fens and their water bodies can be suitable habitats for diverse trichopteran fauna only so long as they possess the requisite environmental conditions, especially with respect to vegetation. Smaller habitat gradients translate into a poorer, less diverse fauna. In the context of the permanence of water bodies and the potential utilisation of Trichoptera in research on fens, it is worth emphasising that appropriate sampling times are as important as the sampling sites. It is more and more often the case that, particularly in macro-scale research, macrobenthos samples are taken just once a year; this saves time and money but may also be in line with specific research aims. When monitoring the aquatic insects of fens, especially the diapausing ones, one should bear in mind that the best period for this is early spring. The results of research carried out in late spring or early summer, particularly in atypically warm seasons, may result in artefacts.

To summarise, our study provides incontrovertible evidence that both spatial and temporal factors affect the faunistic diversity of peatland invertebrates [2]. 

As Boucenna et al. [38] noted, caddisflies are not included in the annexes of Natura 2000 under the European Habitats Directive, and only five representatives of this order have been globally assessed by the International Union for Conservation of Nature [47] so far. This implies serious neglect of the potential of trichopterans, especially in view of the fact that, being amphibiotic insects, they are an ideal group for general habitat evaluations and tracking negative environmental changes. Additionally, as the results of our study have shown, at least at the national level, knowledge gaps can be filled as regards pinpointing reference assemblages or indicator species for calcareous fens and their water bodies. The red-listed *Erotesis baltica* and *Limnephilus fuscinervis* could be additional indicators, even umbrella species, for the most valuable sites of this type—the former for fen pools (or natural small pools) and the latter for the fens themselves. In the case of the Polish population of *Erotesis baltica*, such recommendations have been made before [45] because this species has long been recognised as being particularly valuable for permanently wet open fens [14,35,36]. 

It seems obvious that areas covered by various forms of protection, especially of the highest rank, should be characterised by high values of faunistic indices. In the calcareous fens and their water bodies that we studied, we recorded the greatest richness and diversity of species in the nature reserves and the non-Natura 2000 sites. As with the evaluation of the whole fauna, indices cannot be the only factors determining the natural value of a habitat—this would be too simplistic. Also important in this context are the species level and the faunistic associations in conjunction with habitat characteristics. Thus, 81% of the population of the red-listed *Limnephilus fuscinervis* was recorded at sites in nature reserves. Areas under protection (including nature reserves) in the Republic of Mordovia are also important for high species richness and rare species of Trichoptera [46].

Area conservation turned out to be important for caddisflies, especially for the groups of fens (CCA) and for this fauna as a whole (NMDS). The latter relates in particular to the Natura 2000 areas, the fauna of which is distinctive but at the same time closely associated by faunistic similarity, especially on the habitat-type gradient. Two red-listed species were recorded at Natura 2000 sites. An important aspect regarding the protection of our study sites is that, besides being special areas of conservation, they are also (except for DF-2 and FD-3) important bird areas [47]. The most valuable bird species on calcareous fens also exploit various habitat elements in the water–land system. All conservation measures aimed at achieving the greatest possible structural diversity of this habitat will be of benefit to both groups.

## 5. Conclusions

The aquatic stages of caddisflies of four different fen habitats in south-eastern Poland were studied at different levels of organisation: species, assemblages, ecological indices, and functional groups. Trichopterans showed different responses to environmental variables, both natural and those associated with landscape management, including the area protection type. The most significant environmental factor responsible for the species distribution was habitat persistence. Other key factors were plants (for the fauna of fens and water bodies), the use and management of the surrounding land (pools and ditches), and the level of area protection (fens). Physical and chemical water parameters were not statistically significant. Functional groups proved to be useful in characterising individual fen habitats. Ecological indices alone, although very important, were insufficient for making a full evaluation of fen habitats. The type of area protection also influenced the occurrence of caddisflies, especially in Natura 2000 areas. Principal coordinates analysis allowed the definition of reference trichopteran assemblages, while the IndVal analysis identified indicator species of unique calcareous fens. Caddisflies can be a useful group for monitoring the status of fens (nature evaluation and different disturbances), and according to our results, some protective actions can be recommended, e.g., water level regulation, management of the vegetation modifying succession, and landscape management.

## Figures and Tables

**Figure 1 insects-14-00850-f001:**
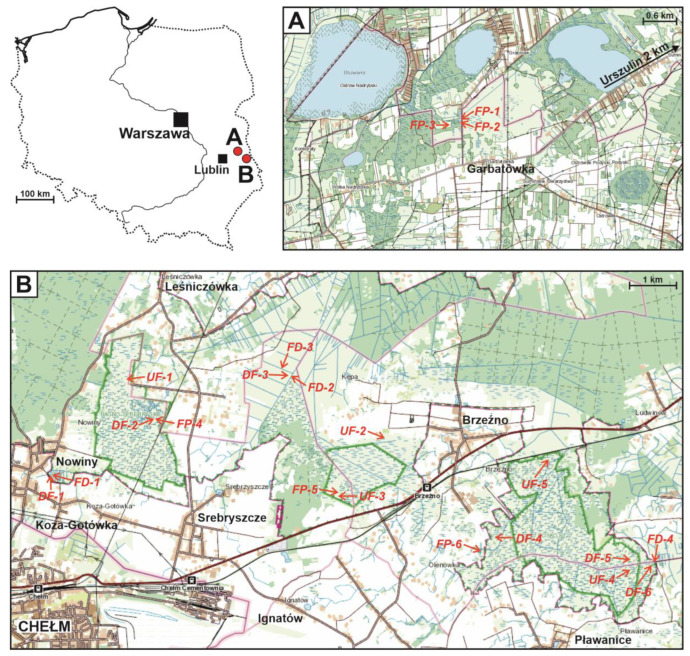
Study area and study sites: A—sites near the village of Urszulin, B—sites near the city of Chełm. UF—undrained fens, DF—drained fens, FD—fen ditches, FP—fen pools. The green lines mark the boundaries of nature reserves. sub-figure (**A**)—western Polesie region, sub-figure (**B**)—Wołyń Polesie region.

**Figure 2 insects-14-00850-f002:**
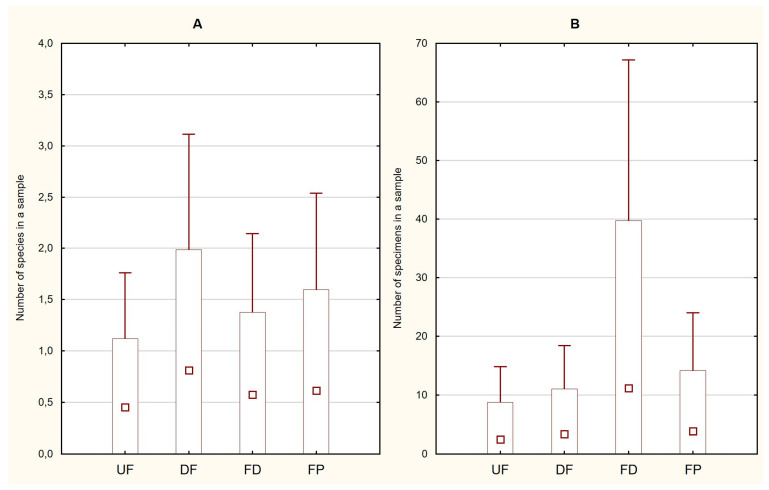
Trichoptera ((**A**)—number of specimens, *p* = 0.7099; (**B**)—species per sample, *p* = 0.7836) in four habitats: UF—undrained fens, DF—drained fens, FD—fen ditches, FP—fen pools. Red square—average value, box—standard deviation, whisker—95% confidence interval.

**Figure 3 insects-14-00850-f003:**
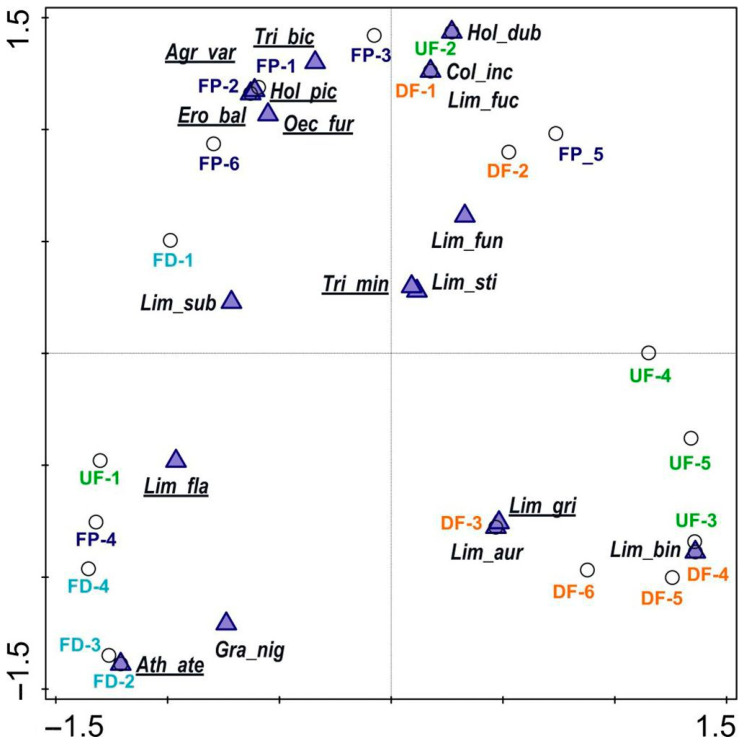
Ordination diagram showing sites vs. caddisfly species with the first two axes of principal coordinates analysis (PCoA): UF—undrained fens, DF—drained fens, FD—fen ditches, FP—fen pools; species are codes as in Table 2; species more sensitive to climate change are underlined, whereas insensitive ones are not.

**Figure 4 insects-14-00850-f004:**
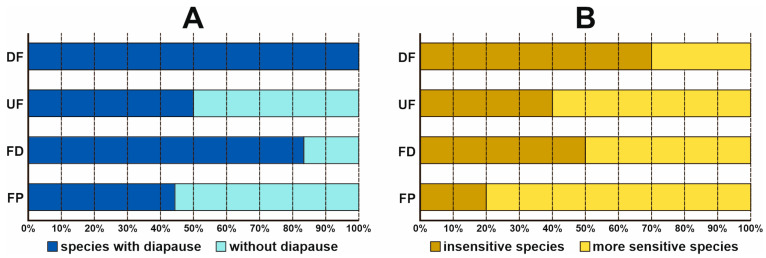
Relative abundance of species in two functional groups of Trichoptera in each habitat type: UF—undrained fens, DF—drained fens, FD—fen ditches, FP—fen pools. (**A**)—drought-resistance strategies; (**B**)—sensitivity to climate change.

**Figure 5 insects-14-00850-f005:**
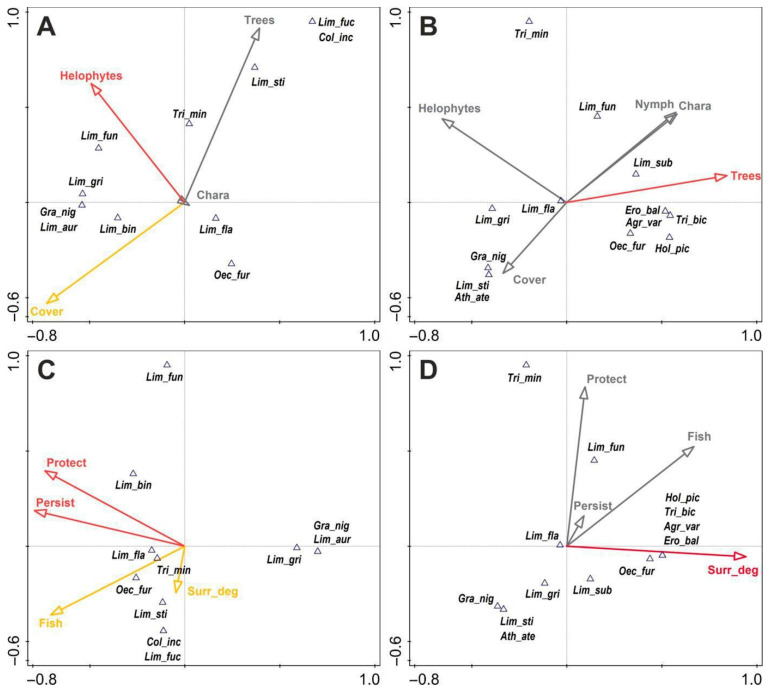
Ordination CCA biplots showing the distribution of Trichoptera species in fens (**A**,**C**) and fen pools and ditches (**B**,**D**) vs. significant (*p* < 0.05—marked in red, *p* = 0.051–0.07—marked in orange) environmental factors: Helophytes—density of emergent vegetation, Nymph—density of floating vegetation, Chara—density of underwater meadows, Cover—overall % coverage of a site by vegetation, Trees—density of woody vegetation, Protect—degree of area protection, Fish—pressure from fish, Surr_deg—extent to which an area is utilised or managed. The taxon codes are as described in Table 2.

**Figure 6 insects-14-00850-f006:**
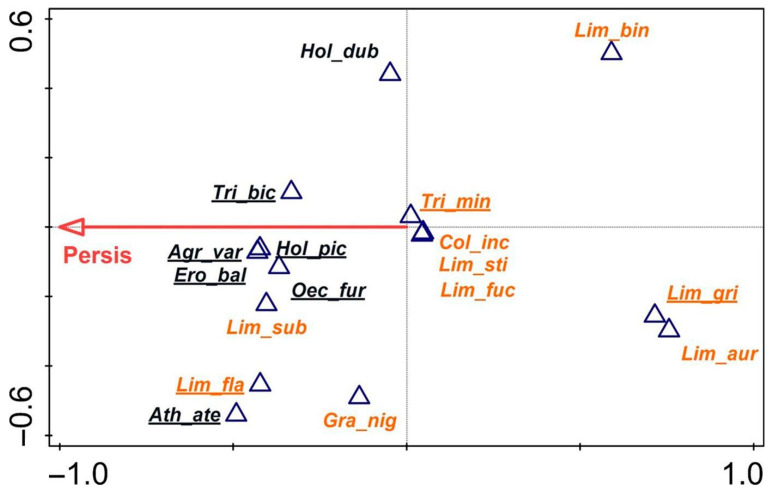
Distance-based redundancy analysis (db-RDA) plot for the trichopteran assemblages of all fen habitats based on the Bray–Curtis distance (dissimilarity). Persistence vector with *p* = 0.043. Species with diapause are shown in orange; those without diapause are shown in black. Δ—species.

**Figure 7 insects-14-00850-f007:**
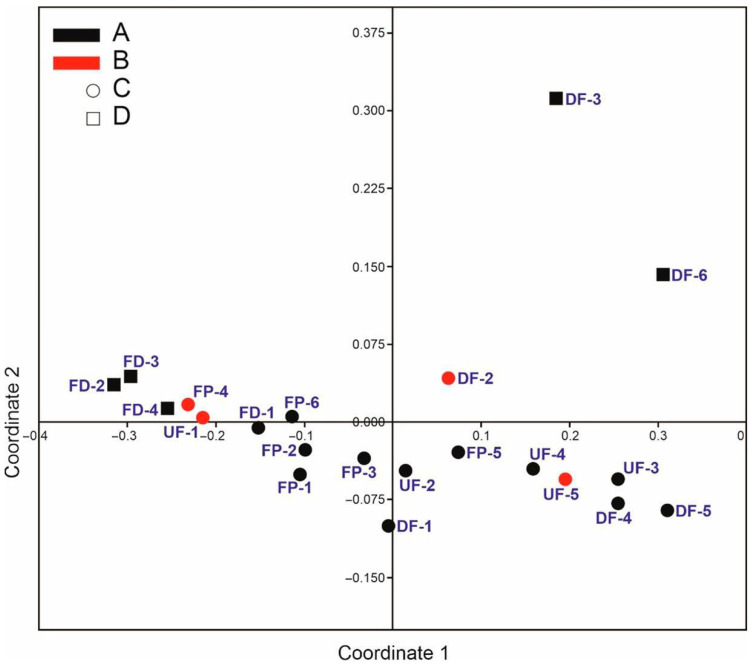
Two-dimensional non-metric multidimensional scaling (NMDS) plot of study sites representing different protection statuses (red—nature reserves, black—unprotected sites, dot—Natura 2000 areas, square—site not in the Natura 2000 network) based on dissimilarities between trichopteran assemblages (Bray–Curtis distance matrix). Stress value = 0.04.

**Table 1 insects-14-00850-t001:** Physical and chemical properties of the water in the various habitats: mean values and range of values. p—probability of comparisons of data from the habitats (Kruskal–Wallis test). UF—undrained fens, DF—drained fens, FD—fen ditches, FP—fen pools. TEMP—temperature, ORP—oxidation reduction potential, DO—dissolved oxygen (mg), EC—electrolytic conductivity, TDS—total dissolved solids.

Data	Site				*p*
	UF	DF	FD	FP	
TEMP [°C]	19.6 (7.27–33.29)	17.5 (7.4–30.7)	15.3 (7.4–26.6)	15.9 (6.39–26.79)	>0.0001
pH	9.00 (706–10.48)	8.50 (68.6–10.69)	8.00 (6.94–10.50)	8.20 (7.16–11.08)	0.1239
ORP [mV]	−172 (−453–206)	−169 (405–220)	−163 (−402–177)	−177 (−472–250)	0.0014
DO [mg·dm^−3^]	3.20 (0.00–9.35)	3.50 (0.0–10.60)	3.79 (0.00–10.25)	2.56 (0.00–10.42)	>0.0001
EC [µS·cm^−1^]	639 (138–1232)	634 (294–2203)	838 (372–3232)	555 (205–1848)	0.0032
TDS [ppm]	322 (84–616)	366 (125–2041)	422 (178–1616)	274 (103–924)	>0.0001

**Table 2 insects-14-00850-t002:** Trichopteran species recorded: occurrence at the study sites and density (ind.⋅m^−2^). UF—undrained fens, DF—drained fens, FD—fen ditches, FP—fen pools. *—Red List species. Ecological indices: species richness (S), abundance (N), the Shannon–Wiener diversity index (H), Buzas–Gibson evenness (E), the Simpson dominance index (D), and the estimate of total species richness (Chao1).

Species	Code	Sites	Habitats			
			UF	DF	FD	FP
*Agrypnia varia* (F.)	Agr_var	FP-2	–	–	–	0.004
*Athripsodes aterrimus* (Stephens, 1836)	Ath_ate	FD-2	–	–	0.006	–
*Colpotaulius incisus* (Curtis, 1834)	Col_inc	DF-1	–	0.028	–	–
*Erotesis baltica* McLachlan, 1877 *	Ero_bal	FP-2	–	–	–	0.004
*Grammotaulius nigropunctatus* (Retzius, 1783)	Gra_nig	DF-3, FD-2, FD-3	–	0.014	0.028	–
*Holocentropus dubius* (Rambur, 1842)	Hol_dub	UF-2	0.005	–	–	–
*Holocentropus picicornis* (Stephens, 1836)	Hol_pic	FP-1, FP-2	–	–	–	0.007
*Limnephilus auricula* Curtis, 1834	Lim_aur	DF-3	–	0.021	–	–
*Limnephilus binotatus* Curtis, 1834	Lim_bin	DF-4	–	0.007	–	–
*Limnephilus flavicornis* (F.)	Lim_fla	all study sites	0.636	0.330	2.861	0.956
*Limnephilus fuscicornis* (Rambur, 1842)	Lim_fuc	DF-1	–	0.014	–	–
*Limnephilus fuscinervis* (Zetterstedt, 1840)*	Lim_fun	UF-3, DF-2, FP-2, FP-4	0.005	0.056	–	0.007
*Limnephilus griseus* (L.)	Lim_gri	DF-2, DF-3, DF-6, FD-3, FP-6	–	0.358	0.006	0.004
*Limnephilus stigma* Curtis, 1834	Lim_sti	DF-1, DF-5, FD-2	–	0.028	0.006	–
*Limnephilus subcentralis* Brauer, 1857	Lim_sub	FD-4, FP-3	–	–	0.006	0.004
*Oecetis furva* (Rambur, 1842)	Oec_fur	UF-1, UF-2, FP-1, FP-2, FP-6	0.015	–	–	0.041
*Triaenodes bicolor* (Curtis, 1834)	Tri_bic	FP-1, FP-2, FP-3	–	–	–	0.022
*Trichostegia minor* (Curtis, 1834)	Tri_min	UF-2, UF-4, DF-1, DF-2, FP-4	0.015	0.014	–	0.007
		S=	5	10	6	10
		N=	132	124	524	285
		H=	0.32	1.50	0.10	0.50
		E=	0.27	0.44	0.18	0.16
		D=	0.88	0.32	0.98	0.82
		Chao1=	6	10	12	11.5

## Data Availability

Data are available from the corresponding author upon reasonable request.

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
