# Peer review of "Caddisflies (Trichoptera) of Protected Calcareous Fen Habitats: Assemblages, Environmental Drivers, Indicator Species, and Conservation Issues"

_insects, 2023, doi:10.3390/insects14110850_

Round 1
Reviewer 1 Report
Comments and Suggestions for Authors
This manuscript describes the results of a faunistic and functional investigation of an important freshwater habitat for an amphibiotic group of organisms with high species and functional diversity. The results and their interpretation are significant contributions for understanding the ecology of calcareous fens, including both those that are in natural condition and those with anthropogenic alteration; furthermore, the attention to effects of dessication suggest some consequences for future climate change.
I look forward to the appearance of the eventually published version of this manuscript.
Comments on the Quality of English LanguageMinor attention to English grammar is needed. (E.g., the Latin word "Trichoptera" is plural and requires a plural verb.)
Author Response
Thank you very much for your kind words and favourable assessment of our article.
Minor attention to English grammar is needed. (E.g., the Latin word "Trichoptera" is plural and requires a plural verb).
We have carefully checked the manuscript and corrected the forms.
Reviewer 2 Report
Comments and Suggestions for Authors
The presented manuscript describes the results of the study of Trichoptera living in the peat fens of Poland. The introduction describes the problems, tasks and goals set by the authors correctly.
There are several comments to the text of the manuscript:
- In the description of the swamps, you must specify the references from which you took the information (description of the fens)
- Why wasn't the material collected in August?
- What types of fish have you accepted as invasive? What did you do with them afterwards if you didn't let them go into the pond?
- In Table 2, give the full Latin name of the species.
- Give the results of calculations of various indices in a separate table or in Table 2.
- For discussion, you need to read this publication (https://dx.doi.org/10.1134/S1995082923010029 ).
However, the manuscript lacks a conclusion and the authors must submit it.
Author Response
Thank you for your tips and comments that contributed to improving the quality of our paper.
In the description of the swamps, you must specify the references from which you took the information (description of the fens)
In the description of the particular sites (types of habitats) we used our own observations and studies (general hydromorphology, vegetation, fish). Therefore – please check – the references were given in the general description of the study site.
Why wasn’t the material collected in August?
Due to the biology of Trichoptera and the habitats we have examined. In late summer, many species are on wings, far from water (and these with diapause are in shelters) and our studies were focused on aquatic stages because only in this case we are sure that the species develop in the studied waters. Imagines, e.g. attracted to light traps, may be migrants from further, distant sites.
What types of fish have you accepted as invasive? What did you do with them afterwards if you didn't let them go into the pond?
In accordance with the law in Poland and throughout the European Union, species known to be invasive are placed on the list and, in accordance with Article 33, cannot be released into the environment (Regulation of the Minister of the Environment of September 9, 2011 on the list of alien plant and animal species which, if released into the natural environment, may threaten native species or natural habitats). In our research it was Perccottus glenii. The caught individuals of this invasive species were killed with Morenicol Lernex Sedation Fluid and then disposed of.
In Table 2, give the full Latin name of the species.
Appropriate changes have been made to Table 2.
Give the results of calculations of various indices in a separate table or in Table 2.
We have added the values of indices to the Table 2.
For discussion, you need to read this publication (https://dx.doi.org/10.1134/S1995082923010029.
We have added the suggested paper to the Discussion section. Since authors examined adult caddisflies (light traps) and unfortunately no fen habitats were present in two protected areas they studied, we referred generally to the studies on Trichoptera in the protected areas.
However, the manuscript lacks a conclusion and the authors must submit it.
We have added the conclusion although according to the guidelines for authors they are not obligatory (therefore they were not provided in the original version of the manuscript).
Reviewer 3 Report
Comments and Suggestions for Authors
Dear authors,
I have no fundamental corrections to your study. However, I would recommend some additions to your manuscript.
1) It is not clear from your manuscript how you identified the species of Trichoptera. You should mention the identification keys that you used. Was DNA analysis required for identification? I am not an expert on Trichoptera, however, to the best of my knowledge, the taxonomy of this group of insect is quite complex.
2) Table 2 List of species. In addition to the author in the species name, you should also include the year. In the list of species this is obligatory.
3) At the first mention in the text of the manuscript you should mention not only the name of the species, but also the author and year. For plants, you have done this, but for Trichoptera it is not everywhere in the text. The text of the manuscript should be checked.
4) The titles of sections 3.1 and 3.2 should be corrected. You mention about “fauna”, however, you are actually study communities of Trichoptera. If you talk about fauna, you should describe the regional aspect in these section, e.g. types of species ranges, origin of regional fauna and so on. In your case, it is incorrect to talk about the influence of environmental factors on "fauna".
Author Response
Thank you for your tips and comments that contributed to improving the quality of our paper.
It is not clear from your manuscript how you identified the species of Trichoptera. You should mention the identification keys that you used. Was DNA analysis required for identification? I am not an expert on Trichoptera, however, to the best of my knowledge, the taxonomy of this group of insect is quite complex.
The first author of the paper (Edyta Buczyńska) is a trichopterologist with over 20 years of experience in this order, therefore we did not see the need to include all widely used keys for identification of trichopteran larvae in Europe. Moreover, species found in the fens we studied are common, well recognized and easy to determine and no DNA analyses were required. However, if the Editors wish, we are ready to provide the list of the keys.
Table 2 List of species. In addition to the author in the species name, you should also include the year. In the list of species this is obligatory.
Appropriate changes have been made to Table 2.
At the first mention in the text of the manuscript you should mention not only the name of the species, but also the author and year. For plants, you have done this, but for Trichoptera it is not everywhere in the text. The text of the manuscript should be checked.
Appropriate changes have been made in the text.
The titles of sections 3.1 and 3.2 should be corrected. You mention about “fauna”, however, you are actually study communities of Trichoptera. If you talk about fauna, you should describe the regional aspect in these section, e.g. types of species ranges, origin of regional fauna and so on. In your case, it is incorrect to talk about the influence of environmental factors on "fauna".
Appropriate changes have been made.
Round 2
Reviewer 2 Report
Comments and Suggestions for Authors
Dear authors. Thank you for answering my questions and comments.
